# Sex-Based Effects on Muscle Oxygenation During Repeated Maximal Intermittent Handgrip Exercise

**DOI:** 10.3390/sports13020042

**Published:** 2025-02-06

**Authors:** Modesto A. Lebron, Justine M. Starling-Smith, Ethan C. Hill, Jeffrey R. Stout, David H. Fukuda

**Affiliations:** 1Institute of Exercise and Rehabilitation Science, University of Central Florida, 4000 Central Florida Blvd, Orlando, FL 32816, USA; modesto.lebron@ucf.edu (M.A.L.); justine.starling-smith@ucf.edu (J.M.S.-S.); ethan.hill@ucf.edu (E.C.H.); jeffrey.stout@ucf.edu (J.R.S.); 2Florida Space Institute, University of Central Florida, 12354 Research Pkwy, Orlando, FL 32826, USA; 3College of Medicine, University of Central Florida, 6850 Lake Nona Blvd, Orlando, FL 32827, USA

**Keywords:** muscle oxygenation, near-infrared spectroscopy, handgrip, sex differences

## Abstract

Background: This investigation aimed to examine sex-based differences in deoxy[heme] (HHb), tissue saturation (StO_2_), and force-deoxygenation ratio (FD) of the forearm flexor muscles during a maximal-effort intermittent fatiguing handgrip protocol. Methods: Thirty-three healthy males (*n* = 15) and females (*n* = 18) completed a fatiguing handgrip protocol consisting of 60 4 s contractions separated by a 1 s rest. Near-infrared spectroscopy was used to measure muscle oxygenation before, during, and after the protocol. Results: Sex differences in HHb (*p* = 0.033) and StO_2_ (*p* = 0.021) were observed with significantly greater values for females (HHb: 110.204 ± 12.626% of baseline; StO_2_: 72.091 ± 5.812%) in comparison to males (HHb: 101.153 ± 12.847% of baseline; StO_2_: 66.978 ± 7.799%). Females (0.199 ± 0.081 AU) also demonstrated significantly (*p* = 0.001) lower FD in comparison to males (0.216 ± 0.094 AU). However, males (b = −0.023 ± 0.008 AU) demonstrated a significantly (*p* < 0.001) greater rate of decline in FD in comparison to females (b = −0.017 ± 0.006 AU). Conclusions: Prior to, during, and after a maximal-effort intermittent fatiguing handgrip fatiguing protocol, males demonstrate significantly lower StO_2_ than females and a faster rate of decline in FD. Moreover, females demonstrate greater HHb values than males when assessed relative to a resting baseline.

## 1. Introduction

Single-effort maximal handgrip strength is commonly used as an indicator of future morbidity, all-cause mortality, and functional decline [1,2]. The significance of handgrip strength as an indicator arises from its close association with overall muscle function and health. Handgrip strength is often utilized as a diagnostic tool to identify muscular atrophy or sarcopenia [3]. Furthermore, decreases in handgrip strength have been associated with an increase in activities of daily living disability [4], potentially leading to a decrease in quality of life [5]. Declining handgrip strength has also been linked to metabolic [6], cardiovascular [7], and cognitive health [8], making it a comprehensive marker of overall well-being.

Although single-effort maximal handgrip strength provides insights for overall health, sustained or repeated handgrip strength is a requirement for many daily activities, sports, and occupations. For example, assembly-line laborers being observed over 25–45 min sessions were engaged in a gripping action for an average of 88% of the total time on task [9]. Dependent on intensity and the nature of intermittency, repetitive and forceful handgrip exertions may lead to muscular fatigue. Fatigue resistance has been observed to be sex-dependent, with females being more fatigue resistant than males at different exercise intensities [10] and muscle groups [11,12]. This fatigue resistance has been attributed to several physiological differences, including a relatively higher composition of type I muscle fibers [13], greater muscular capillarization [13], and more efficient mitochondrial function [14] in females compared to males, all of which may influence measures of muscle oxygenation.

During exercise, muscle oxygenation responses can be measured using near-infrared spectroscopy (NIRS) [15]. More specifically, NIRS can assess changes in oxygenated (oxy[heme], O_2_Hb) and deoxygenated hemoglobin (deoxy[heme], HHb), which provide insight into oxygen delivery and uptake within active tissues, reflecting potential physiological differences between populations. Understandably, sex-based differences have been observed in muscle oxygenation and deoxygenation for the lower limbs [16,17,18,19] and upper limbs [20,21]. When exercising, muscle oxygenation may contribute to fatigue or a loss in force output over time. For example, completing single-leg knee extensions in hyperoxia (70% O_2_ in balance with N_2_) results in higher muscle oxygenation in the vastus lateralis and improved exercise tolerance (i.e., greater time to exhaustion) [22]. Moreover, previous research has demonstrated the advantage in fatigue resistance for females, in comparison to males, to be eliminated when exercising under ischemic conditions [23]. Collectively, NIRS may be used as an index of physiological differences between males and females in a non-invasive manner.

When completing intermittent isometric knee extensions to exhaustion, 10% above relative critical intensity, males demonstrated a greater increase in HHb and a greater decrease in tissue oxygenation index compared to females [16]. Furthermore, when completing 10 maximal-effort isokinetic knee extensions at 60°·s^−1^, males showed a greater decrease (18.9% ± 14.3%) in tissue saturation (StO_2_) compared to females (6.6% ± 4.8%) [19]. Similar findings have been reported during handgrip exercise. When completing a passive vascular occlusion test (VOT) before and after a sustained handgrip task (SHG), males (pre VOT: b = −0.208, post VOT: b = −0.079, SHG: b = −0.070) demonstrated a more rapid rate of oxygen desaturation when compared to females (pre VOT: b = −0.123, post VOT: b = −0.070, SHG: b = −0.015), even when being matched for strength [20]. Additionally, when completing sustained handgrip time to exhaustion bouts at intensities equal to 20, 40, 60, and 80% of maximal voluntary contraction (MVC), males, again, demonstrated a more rapid rate of desaturation, with the difference being more prevalent in the higher intensities [21].

Although several previous studies have observed oxygenation and deoxygenation during submaximal handgrip exercise [20,21,24], fewer studies have examined muscle oxygen kinetics during intermittent maximal-effort exercise, with no research to our knowledge comparing sex-based differences. During a 6 min repeated maximal-effort handgrip assessment in males, Nakada et al. [25] observed a significant correlation between time to highest HHb value and time to 70% MVC, as well as exponential rate of decrement (*k*). However, the oxygenation response is contraction interval dependent, and the time to reach the highest value for HHb is significantly greater with longer intervals [26]. Thus, muscle oxygenation may influence force production capabilities, and utilizing a shorter rest interval between contractions may accelerate the observed oxygenation response.

A better understanding of sex-based differences in muscle oxygenation during handgrip exercises may provide implications regarding the impacts of fatiguing handgrip exercise. Therefore, the purpose of this study was to examine sex-specific differences in the changes in StO_2_, HHb, and force–deoxygenation ratio (FD) of the forearm flexor muscles during a maximal-effort intermittent fatiguing handgrip protocol. Based on previous research [24], we hypothesized that there would be no difference in HHb between males and females before, during, or after the intermittent fatiguing protocol. Additionally, based on findings from Mantooth and colleagues [21], we hypothesized that females will have greater StO_2_ compared to males, with males demonstrating a higher rate of decrease in FD during the intermittent fatiguing protocol.

## 2. Materials and Methods

### 2.1. Experimental Design

A randomized cross-over design was used to determine the effects that sex may have on muscle oxygenation and FD. Participants reported to the laboratory on three separate occasions: an informed consent visit (V1), a familiarization visit (V2), and an experimental visit (V3) randomly assigned to different times of day (9 a.m., 2 p.m., or 7 p.m. EST). During the experimental visit, participants completed a maximal-effort intermittent handgrip fatiguing protocol consisting of 60 4 s maximum effort isometric contractions while simultaneously measuring muscle oxygenation with an NIRS device. All procedures were approved by the University’s Institutional Review Board (STUDY00004620).

### 2.2. Participants

Thirty-three healthy males and females (Table 1) between the ages of 18 and 55 with no reported cardiovascular, metabolic, or renal diseases volunteered to participate in this study. Additionally, participants reported no physical limitations preventing maximal-effort handgrip. This partial sample was extracted from a larger study with a separate purpose [27], assessing MVC peak force between males and females at different times of day. A total of 8 individuals (4 males, 4 females) were excluded from the original sample for our final analysis as a result of insufficient NIRS data. Prior to beginning the study, all participants received a written and verbal explanation of experimental protocols and provided a signed, written informed consent. Following the written consent, participants completed a physical activity readiness questionnaire and a demographics questionnaire. After completing all questionnaires, an ultrasound device was used to assess adipose tissue thickness of the forearm at the site of the NIRS device placement. Self-reported dominant hand and chair position were established and recorded to be replicated during familiarization and the experimental visit. Participants were instructed to arrive to the familiarization visit and the experimental visit well hydrated, 2 h fasted, and refraining from exercise and caffeine consumption 12 h prior.

### 2.3. Familiarization Visit

During the familiarization visit, height, weight, and body composition were recorded. Height and weight were measured using a scale and stadiometer, where body composition was assessed using multi-frequency bioelectrical impedance analysis (InBody 770, Biospace Co., Ltd., Seoul, Republic of Korea). For body composition, participants were asked to remove shoes, socks, and jewelry prior to assessment. Familiarization of the intermittent handgrip fatiguing protocol follows the same procedures outlined in the experimental visit below.

### 2.4. Experimental Visit

Upon arrival to the experimental visit, participants had an NIRS device secured above the flexor digitorum superficialis muscle. Participants were instructed to remain seated and still for 2 min to allow the NIRS device to stabilize. Participants then completed one MVC with the handgrip dynamometer (TSD121C, BioPac Systems, Inc.; Goleta, CA, USA), used to determine relative intensities for the standardized warm-up. Following the MVC, participants completed a set of 10 repetitions at 25% MVC, a set of 6 repetitions at 50% MVC, and a final set of 10 repetitions at 25% MVC. Each set during the standardized warm-up was separated by 30 s. Following the warm-up, participants completed 3 MVCs with one minute of rest between each effort. One minute after the final MVC, participants completed the intermittent handgrip fatiguing protocol, consisting of 60 4 s maximum effort isometric contractions with 1 s of relaxation between contractions. An iPad (Apple Inc., Cupertino, CA, USA) was positioned in front of the participant, and a timer application was used to provide visual and auditory cues to the participant dictating phases of contraction and relaxation.

For the warm-up and the intermittent handgrip fatiguing protocol, participants were in a seated position with their back flat against the chair and their hips as far back in the chair as possible. Their feet were flat on the floor with the knees, hips, and elbows flexed at 90° angles, with the arm supported by an adjustable armrest. They firmly grasped the dynamometer in their dominant hand and kept their wrist in a neutral position for the duration of the protocol. Each visit was separated by at least 24 h.

### 2.5. Near-Infrared Spectroscopy (NIRS)

During the experimental visit, muscle oxygenation was monitored using a portable, continuous, dual-wavelength (760 and 850 nm) NIRS device (PortaLite, Artinis Medical Systems, Arnhem, The Netherlands). This device records data using three optode channels (Tx1, Tx2, Tx3) with optode distances at 30, 35, and 40 mm. The flexor digitorum superficialis muscle was gently palpated to identify the muscle belly and marked with a permanent marker. The NIRS device was placed over this mark and secured in place with an adhesive bandage and covered with darkly colored athletic tape to prevent exogenous light contamination. This device provides relative measurements of O_2_Hb, HHb, total[heme] (THb), and StO_2_, performed in accordance with the modified Beer–Lambert law. Data were recorded at a sampling rate of 10 Hz using a differential pathlength factor of 4 for all participants, which reflects the average value for DPF in the human forearm [28]. Throughout the exercise bouts, the investigator visually monitored, in real time, the percentage of light reaching the photodiode and the signal fit factor to ensure signal quality. HHb is recorded as an arbitrary unit (AU) and was reported from the Tx2 channel with an optode distance of 35 mm. StO_2_ is recorded as a percentage and was reported from the combination of data from all three optode distances.

Prior to any handgrip contractions, participants were instructed to remain seated and still for 2 min to allow the NIRS device to stabilize. The last 30 s of the 2 min stabilization period were averaged for each variable (i.e., HHb, StO_2_) and used as a measure of baseline (HHb = 46.47 ± 0.45 AU; StO_2_ = 71.17 ± 0.50%). In the case of the variables failing to stabilize during the last 30 s (*n* = 4), a separate stable range of 30 s was used. With a total of 60 contractions, the intermittent handgrip fatiguing protocol spanned 5 min (300 s) in total, with muscle oxygenation being measured continuously throughout. To assist with the analysis of changes in HHb and StO_2_, all data were averaged over 30 s time bands (i.e., ranges). Range 1 was 0–30 s, range 2 was 31–60 s, range 3 was 61–90 s, and so on until range 10 was 270–300 s. Each individual handgrip contraction lasted 4 s followed by 1 s of rest. As such, each range corresponded to six consecutive contractions and the HHb and StO_2_ values measured within that range. Additionally, the 30 s immediately before the initiation of the protocol (PRE) and the 30 s immediately following the completion of the protocol (POST) were also analyzed. HHb is reported as a percentage of baseline [25,29], whereas StO_2_ is expressed as a percentage and calculated as ([O_2_Hb] × ([O_2_Hb] + [HHb])^−1^) × 100. FD for each range was calculated as ((average handgrip peak force × body mass^−1^) × average HHb^−1^) × 100, over the respective 30 s range. This calculation aligns with previous methodologies used by Paradis-Deschênes and colleagues [30], calculating what they define as metabolic efficiency. However, we additionally normalized handgrip peak force to body mass. Lastly, the rate of decrease (slope, *b*) in FD was calculated for each participant.

### 2.6. Statistical Analysis

A series of two-way mixed-factorial ANOVAs (range × sex) were used to examine differences in all dependent variables (HHb, StO_2_, ME). In the case of significance, Holm post hoc analyses were conducted. Handgrip peak force obtained from baseline MVC was utilized as a covariate when analyzing StO_2_. A one-way ANOVA was used to examine difference in alterations of FD over time (slope, *b*) between males and females. Significance was considered at a *p* value ≤ 0.05. All data are reported as mean ± SD. All statistical analysis were conducted using JASP (0.16.2.0).

## 3. Results

### 3.1. Deoxy[heme]

No range × sex interaction was observed for HHb (F(2.68,83.05) = 0.847, *p* = 0.461, η_p_^2^ = 0.027) (Figure 1). There was, however, a main effect for range (F(2.68,83.05) = 7.211, *p* < 0.001, η_p_^2^ = 0.189). Post hoc analysis identified PRE to be significantly lower than all following ranges (*p* < 0.002), except range 1 and POST (*p* > 0.05). Additionally, range 1 was significantly less than ranges 3–10 (*p* < 0.05), with no significant differences between any two ranges between 2 and 10 (*p* > 0.999). Lastly, a main effect of sex (F(1,31) = 4.991, *p* = 0.033, η_p_^2^ = 0.139) was observed, with males demonstrating significantly less HHb than females.

### 3.2. Tissue Oxygen Saturation

Similarly to HHB, no range × sex interaction was observed for StO_2_ (F(2.39,74.19) = 0.766, *p* = 0.490, η_p_^2^ = 0.024) (Figure 2). There was, however, a main effect for range (F(2.39,74.19) = 4.752, *p* = 0.008, η_p_^2^ = 0.133). Post hoc analysis identified baseline StO_2_ was significantly (*p* = 0.039) greater than StO_2_ for range 2, with no difference in StO_2_ between baseline and repetition ranges 1, 3–10, or POST. Additionally, PRE StO_2_ was significantly greater (*p* < 0.05) than ranges 2–10, but not POST. StO2 at POST was significantly greater than ranges 2–4 and 8 (*p* < 0.05). Furthermore, a main effect of sex (F(1,31) = 5.930, *p* = 0.021, η_p_^2^ = 0.161) was observed, with males demonstrating significantly lower mean StO_2_ than females. Peak force from baseline MVC did not significantly influence this relationship (F(1,30) = 0.722 *p* = 0.402, η_p_^2^ = 0.023).

### 3.3. Force-Deoxygenation Ratio

A range × sex interaction was observed for FD (F(1.96,60.86) = 3.242, *p* = 0.047, η_p_^2^ = 0.095) (Figure 3). Post hoc analysis revealed that FD did not differ significantly between males and females when comparing between the same range. For both males and females, range 1 was greater than ranges 2–10, range 2 was greater than 3–10, and range 3 was greater than 5–10. Range 4 was greater than ranges 6–10 and ranges 8–10 for males and females, respectively. For the purpose of brevity, male and female comparisons outside of specific ranges are not reported. However, the interaction is supported by the significant difference observed when analyzing slope. Males (b = −0.023 ± 0.008) demonstrated a significantly greater rate of decline in FD (F(1,31) = 5.106, *p* = 0.031, η_p_^2^ = 0.141), in comparison to females (b = −0.017 ± 0.006) (Figure 4).

A main effect of range was also observed for FD (F(1.96,60.86) = 201.838, *p* < 0.001, η_p_^2^ = 0.867). Post hoc analysis revealed ranges 1–3 to be significantly greater than all subsequent ranges to follow, respectively (*p* < 0.01). Furthermore, range 4 was observed to be significantly greater than ranges 6–10 (*p* < 0.01), range 5 significantly greater than ranges 7–10 (*p* < 0.011), and range 6 to be significantly greater than ranges 9 and 10 (*p* < 0.035). No main effect of sex was observed for FD (F(1,31) = 0.907, *p* = 0.348, η_p_^2^ = 0.028).

## 4. Discussion

The aim of the current research study was to identify potential differences in muscle oxygenation between males and females during an intermittent maximal-effort handgrip fatiguing protocol. Additionally, we attempted to quantify FD and observe the relationship between force output and deoxygenation. Our main findings were the following: (1) when normalized to a respective baseline, sex had a significant influence on HHb; (2) StO_2_ significantly decreases within the first 30 s of the onset of maximal-effort intermittent handgrip exercise but remains relatively unchanged until the termination of exercise; (3) males demonstrated a significantly lower percentage of tissue saturation when compared to females; (4) males demonstrated a significantly steeper decrease in FD in comparison to females.

### 4.1. Muscle Oxygenation

In support of previous research in the upper extremities [24,25,26], this study demonstrated a significant increase in HHb following the onset of exercise. Additionally, increases in HHb, as a percentage of baseline, demonstrated a similar response to that observed in previous research [25,26]. Prior to the investigation, we hypothesized observing no difference in HHb between males and females before, during, or after the intermittent fatiguing protocol. However, in contrast to our hypothesis and previous findings [24], females in this study demonstrated greater HHb than males at rest, with the difference maintained throughout the maximal-effort intermittent handgrip fatiguing protocol. Although not maximal effort, Keller et al. [24] observed no differences in HHb increases between males and females when completing isometric handgrip contractions to failure at 25% MVC. Keller and colleagues [24] normalized HHb values by means of a vascular occlusion test, which allows the accurate depiction of peak values. This may be a contributing factor when comparing to our findings, as HHb was normalized to a resting baseline value. Moreover, females have demonstrated greater proportions of type I fibers when compared to males [31]. HHb reflects microvascular O_2_ extraction; therefore, the greater recruitment of primarily oxidative fibers for females may partially explain the difference in resting and exercise HHb values. However, the methodological design of this study prevents the ability to identify the exact physiological mechanisms responsible for this observation. Lastly, conflicting evidence is present when observing changes in HHb in the lower limbs. Ansdell and colleagues [16] observed females to demonstrate a significantly smaller increase in HHb when completing intermittent isometric knee extensions to task failure, assuming differences in muscle oxygenation between upper and lower limbs. A similar response was observed when males and females completed ten 10 s sprints under normoxic (21% F_I_O_2_) and acute hypoxic (13% F_I_O_2_) conditions, where the magnitudes of HHb increases were significantly less for females when compared to males, regardless of condition [32].

Previous research has observed sex-based differences in tissue oxygenation of the forearm; however, these studies implemented submaximal intensities to failure [21,24] and/or passive vascular occlusion tests [20]. The findings observed during submaximal exercise and/or passive occlusive protocols may exhibit limited translation of the fatigue responses related to NIRS measurements during maximal exercise. This was the first study, to our knowledge, to identify sex-based differences in tissue saturation dynamics during an intermittent maximal-effort handgrip fatiguing protocol. Prior to the investigation, we hypothesized observing females to demonstrate greater StO_2_ when compared to males. Although the influence of sex and range is independent, our hypothesis was confirmed and showed females to have significantly higher tissue saturation than males when averaged across ranges, supporting the findings from previously mentioned studies [20,21,24]. With StO_2_ calculated as ([O_2_Hb] × ([O_2_Hb] + [HHb])^−1^) × 100, it reflects the relationship between oxygen supply and consumption. With females demonstrating greater HHb, the difference in tissue saturation supports the greater capacities for O_2_Hb supply and microvascular extraction in females. As mentioned previously, females have been observed to have greater muscular capillary density [13] and more efficient mitochondrial function [14]. Furthermore, when matched for strength, the difference in StO_2_ between females and males still exists, providing additional support that the difference is related to physiological function associated with oxygen supply, fiber composition, and mitochondrial function [20]. However, these differences were noted at rest, and exercise at high intensities (i.e., sustained maximal force production) may cause increases in intramuscular pressure [33] that could ultimately cause intramuscular occlusion.

Prior research [34] has identified that participants categorized as having high strength tend to demonstrate occlusion at a lower percentage of their MVC (51.5%) compared to those categorized as having low strength (75.5%). Moreover, the absolute force producing occlusion did not differ between high- and low-strength groups [34]. Additionally, McNeil and colleagues [35] identified tissue saturation to be influenced by exercise intensity, with more rapid decreases at higher relative intensities. This is in support of findings by Demura and colleagues [26], who identified that decreasing the time between handgrip intervals (i.e., increasing relative intensity) results in an accelerated muscle oxygenation response. However, it is necessary to mention that because the protocol in the present study is intermittent, the 1 s rest may result in a lack of complete occlusion [26] and may not be influenced by force production. Taken together, a combination of (1) the difference in absolute force produced by males and females and (2) physiological differences in muscle fiber composition, capillary density, and mitochondrial efficiency may be influencing factors in the differences in muscle oxygenation when completing maximal-effort contractions. However, with peak force demonstrating no significant impact on the sex-based difference in tissue saturation, the differences when completing intermittent exercise may favor the latter reasoning.

### 4.2. Force-Deoxygenation Ratio

In an effort to observe the dynamic relationship between force output and oxygen extraction, FD was calculated in line with methodologies described by Paradis-Deschênes et al. [30]; however, force values were normalized to body mass. FD was observed to decrease progressively to range 3, or repetitions 13–18, where FD appeared to reach a nadir. Additionally, similar to the outcome observed by Keller and colleagues [24] during submaximal handgrip, our study observed males to demonstrate a significantly greater rate of decrease for FD. However, as described previously, HHb remained relatively unchanged between ranges 2 and 10, indicating that the decrease in FD was largely the result of a reduction in average peak force across ranges without changes in HHb. As such, HHb may not fully reflect metabolic stress related to muscular fatigue when performing intermittent maximal-effort handgrips.

### 4.3. Limitations

Although this is the first study, to our knowledge, to assess the effects of sex on muscle oxygenation and FD during maximal-effort intermittent handgrip exercise, this study is not without limitations. First, although there is research to support the methodology used to calculate NIRS variables relative to baseline [25], other studies have utilized a vascular occlusion test to normalize raw NIRS signals [24]. In doing so, signals obtained during an intervention could be compared to individualized minimum and maximum values. Second, participants in this study were not required to have any minimal physical activity levels. As such, the results reported may be accurate for the general population; however, potential differences may exist for individuals with specific training backgrounds. Lastly, although the timeframe of exercise (5 min) typically corresponds to more oxidative contributions, without measures such as lactate concentrations or heart rate values, inferences regarding the exact role of energy systems during this protocol cannot be made. Future research may aim to identify population or training background differences in muscle oxygenation during intermittent maximal-effort handgrip exercise.

## 5. Conclusions

This study evaluated forearm muscle oxygenation before, during, and after an intermittent handgrip fatiguing protocol, with comparisons made between sexes. Additionally, FD was assessed during the protocol to characterize the dynamic relationship between force output and microvascular oxygen extraction over time. Males were observed to demonstrate significantly lower StO_2_ than females, as well as a faster rate of decline in FD. Furthermore, females demonstrated greater HHb values at before, during, and after the protocol. Lastly, peak force from baseline MVC provided no influence on the relationship between sex-based differences in tissue saturation. Taken together, this further suggests potential physiological differences in the capacities for O_2_Hb supply and microvascular oxygen extraction between males and females being the driving factor for differences in muscle oxygenation, with intramuscular pressure as a result of force production potentially being less influential during intermittent exercise. However, further research is needed to determine the specific contribution or influence of factors such as muscle fiber type, mitochondrial function, and capillary density on muscle oxygenation differences between males and females.

## Figures and Tables

**Figure 1 sports-13-00042-f001:**
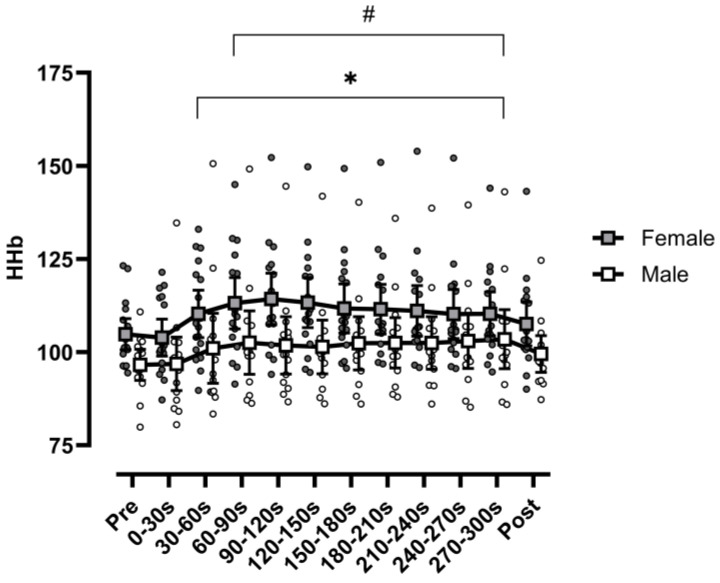
Deoxy[heme] value relative to baseline (%) immediately before the protocol (PRE), at 10 ranges of 30 s intervals during the protocol (ranges 1–10) and immediately after the protocol (POST). Data are separated by sex and presented as individual participants (circles) and means (squares), with error bars reflecting standard error. Filled shapes reflect females, while hollow shapes reflect males. Females had significantly greater HHb when collapsed across time. (*) Ranges 2–10 were significantly greater than PRE when collapsed across sex; (#) ranges 3–10 were significantly greater than range 1 when collapsed across sex.

**Figure 2 sports-13-00042-f002:**
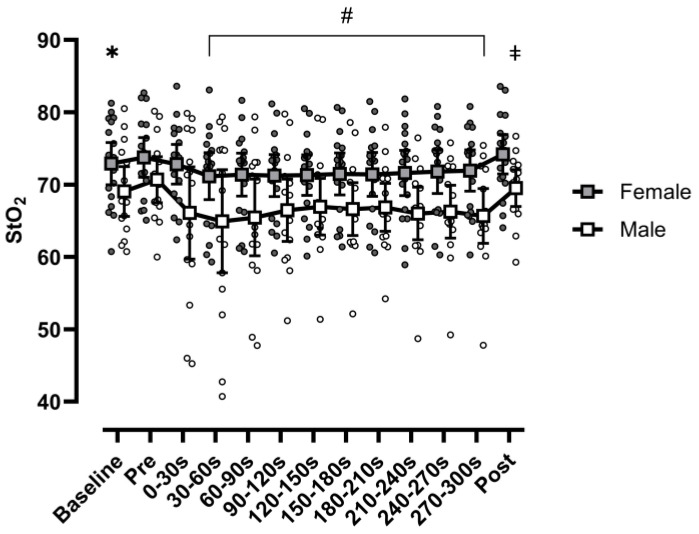
Tissue saturation index, calculated as ([O_2_Hb] × ([O_2_Hb] + [HHb])^−1^) × 100, at baseline, immediately before the protocol (PRE), at 10 ranges of 30 s intervals during the protocol (ranges 1–10), and immediately after the protocol (POST). Data are presented as individual participants (circles) and means (squares), with error bars reflecting standard error. Filled shapes reflect females, while hollow shapes reflect males. Females had significantly greater StO_2_ when collapsed across time. (*) Baseline significantly greater than range 2 when collapsed across sex; (#) ranges 2–10 significantly less than PRE when collapsed across sex; (‡) post significantly greater than ranges 2–4 and 8 when collapsed across sex.

**Figure 3 sports-13-00042-f003:**
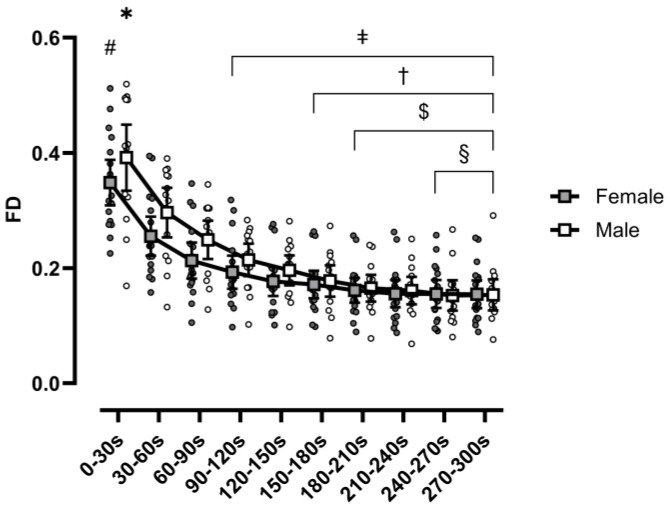
Force-deoxygenation ratio, calculated as ((average handgrip peak force × bodyweight^−1^) × average HHb^−1^) × 100, at 10 ranges of 30 s intervals during the protocol (ranges 1–10). Data are presented as individual participants (circles) and means (squares), with error bars reflecting standard error. Filled shapes reflect females, while hollow shapes reflect males. (#) Range 1 for females significantly greater than all following female ranges; (*) Range 1 for males significantly greater than all following males ranges; (‡) Ranges 4–10 significantly less than ranges 1–3 when collapsed across sex; (†) ranges 6–10 significantly less than range 4 when collapsed across sex; ($) ranges 7–10 significantly less than range 5 when collapsed across sex; (§) ranges 9–10 significantly less than range 6 when collapsed across sex.

**Figure 4 sports-13-00042-f004:**
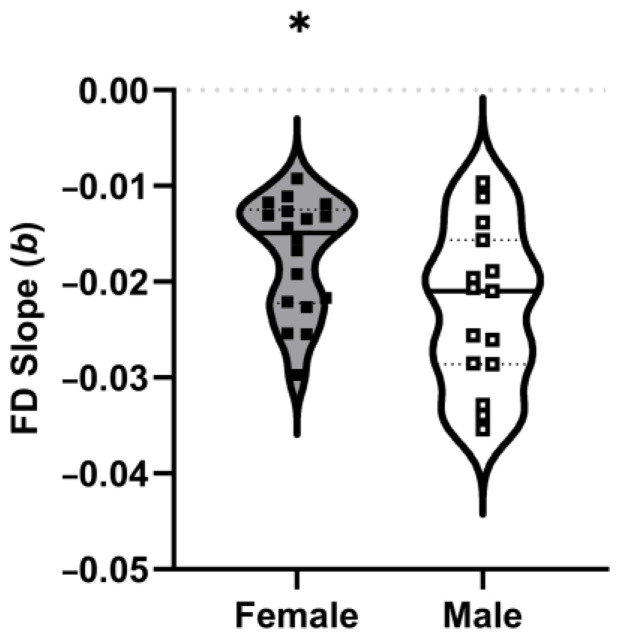
Rate of decrease in force-deoxygenation ratio, calculated as slope (*b*), at 10 ranges of 30 s intervals during the protocol (ranges 1–10), for males and females. The data are presented as a violin plot with individual participant slopes (squares), group means (crossbar), and upper and lower quartiles. Filled shapes reflect females, while hollow shapes reflect males. (*) Significantly less than males.

**Table 1 sports-13-00042-t001:** Participant demographics.

	Total		Age (y)	Height (cm)	Body Mass (kg)	Bodyfat (%)	Adipose Tissue Thickness (cm)	Peak Force (kgf)
Males	(*n* = 15)	Mean ± SD	23.5 ± 6.6	175.2 ± 6.9	84.6 ± 16.7	20.3 ± 7.5	0.4 ± 0.2	39.0 ± 7.3
		Range	19–44	162.5–190.5	53.3–124.4	11.2–35.9	0.3–1.0	22.1–53.6
Females	(*n* = 18)	Mean ± SD	21.6 ± 1.7	165.1 ± 7.3	70.6 ± 16.1	30.5 ± 3.4	0.7 ± 0.2	29.8 ± 5.6
		Range	19–26	153.7–176.3	46.8–107.3	19.0–46.1	0.3–1.2	21.8–41.5

## Data Availability

The data supporting the conclusions of this article can be made available by the authors upon reasonable request.

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
