# Peer review of "Sex-Based Effects on Muscle Oxygenation During Repeated Maximal Intermittent Handgrip Exercise"

_sports, 2025, doi:10.3390/sports13020042_

Round 1
Reviewer 1 Report
Comments and Suggestions for Authors
Dear authors, I would like to congratulate you on your article.
On a general level your work meets all specifications and is of interest.
I would like to make some small suggestions to your work:
- The authors often refer to their work as a project, but I think it would be more appropriate to use work, research, study, etc. as a synonym. The word project can be a bit confusing.
- The authors clearly present the objectives and the basis for their hypotheses, which is appreciated. Please refer to their hypotheses, whether or not they are fulfilled, in the discussion section and not only to the objectives.
- I think the more appropriate word to refer to the number of repetitions would be set rather than range.
- I would recommend that the authors first perform a multiple regression to establish the explanatory nature of the different variables on a dependent variable. This could allow them to make inferences in their work. However, this is not the objective of the present study, since they are only trying to determine possible differences between sexes.
- Due to the sample size of this study, did the authors perform a normality and homogeneity analysis of the variables?
Author Response
- The authors often refer to their work as a project, but I think it would be more appropriate to use work, research, study, etc. as a synonym. The word project can be a bit confusing.
- We edited the manuscript to refer to this work as a study, rather than a project.
- The authors clearly present the objectives and the basis for their hypotheses, which is appreciated. Please refer to their hypotheses, whether or not they are fulfilled, in the discussion section and not only to the objectives.
- References of our hypotheses were added to our discussion – Lines 283-285 and 312-314.
- I think the more appropriate word to refer to the number of repetitions would be set rather than range.
- Although the reviewer suggested using the term sets to describe each 30 second range, we feel that the term “set” may be interpreted as having designated breaks between each range as "sets" within resistance training does.
- Methods were edited to provide a more clear and concise description of the ranges and data to better communicate the data – Lines 177-186
- I would recommend that the authors first perform a multiple regression to establish the explanatory nature of the different variables on a dependent variable. This could allow them to make inferences in their work. However, this is not the objective of the present study, since they are only trying to determine possible differences between sexes.
- We recognize the reviewer’s suggestion of performing a multiple regression but agree with the reviewer who states it is not the objective of the study. However, we intend to consider and incorporate this into future study designs.
- Due to the sample size of this study, did the authors perform a normality and homogeneity analysis of the variables?
- An analysis of normality and homogeneity has been conducted.
- Our analysis demonstrated a violation of normality of HHb (11/12 time points) and a violation of homogeneity for StO2 (3/13 time points); however, we believe an ANOVA is robust enough to provide valid results and Q-Q plot of residuals for StO2 demonstrated relatively normal variance patterns.
- A log transformation was conducted and considered; however, this provided similar outcomes for the respective variables.
- Though normality issues were present for HHb, FD is reflective of the relationship between HHb and relative strength and no normality concerns were present.
Additionally:
- FD replaced ME throughout manuscript.
- Body mass replaced weight throughout manuscript.
Reviewer 2 Report
Comments and Suggestions for Authors
The authors decided to investigate an interesting, current problem. The energy supply of loads of different intensity and type is one of the key issues of physical activity. The choice of topic is justified.
Even though the sample and the methods used already foreshadow a strong limitation of the results, the work is methodologically valuable.
The NIRS method used is adequate for estimating tissue oxygenation. The study of deoxy hemoglobin and tissue saturation is currently a widespread and valuable methodology.
The use of body weight to normalize the data in this case is not justified, perhaps it would have brought us closer to the solution to take advantage of the possibility of bioimpedance measurement (InBody 770 gives approximate estimates of muscle mass distribution), although we should treat this with great caution.
The reason for the gender comparison cannot be disputed, but the manuscript does not reveal the physiological basis of the authors' intentions. Perhaps this is also why the last sentence of the abstract is a theoretical statement that needs to be disputed and explained.
There are no objections to the preparation of the research, careful planning preceded the studies.
Unfortunately, the communication of basic data is very limited and it would be advisable to supplement them.
Since NIRS techniques primarily allow the study of changes, it would have been good to know, for example, the resting laboratory values of the studied properties (Hb, HHB, satO2, etc.).
The range values of the dynamometric data could provide information about the properties of the sample examined.
What is interesting at first glance is the difference between sex of the rest values of HHb, the stO2 to be treated as a base. For ME, this means phase Range 1. These differences can also be felt during lactation, consistently in the case of HHb and satO2, and in the case of FD the evolution of the difference is very interesting.
In explaining the results, the authors have taken a trajectory in which the explanation of the measured results inevitably leads to theoretical findings.
Based on the sample and method used, it cannot be stated that there are physiological differences in the oxygen supply systems of the two genders. This may be partly true, but stronger contributors should also be assumed, such as ST%, individual capillarization status, relationship between individual muscle mass and physical performance, etc.
It is worth mentioning that the applied loads do not specifically count on aerobic sources, it is difficult to give a correct answer regarding the role of energy supply systems in these contractions (see lactate concentration, HR values).
I consider the thesis remarkable, but recommend it for revision.
Author Response
- The use of body weight to normalize the data in this case is not justified, perhaps it would have brought us closer to the solution to take advantage of the possibility of bioimpedance measurement (InBody 770 gives approximate estimates of muscle mass distribution), although we should treat this with great caution.
- The use of body mass to normalize strength allows for a practical and replicable way to differentiate the strength differences on average between males and females. Moreover, BIA has been shown to be inconsistent and may overestimate or underestimate fat free mass. As such, potential differences may lead to an inability to compare our data with subsequent investigations or the accumulation of normative data.
- The reason for the gender comparison cannot be disputed, but the manuscript does not reveal the physiological basis of the authors' intentions. Perhaps this is also why the last sentence of the abstract is a theoretical statement that needs to be disputed and explained.
- Alterations to the introduction were made to the introduction to provide further physiological basis regarding the intention of the study and value of its outcomes – Lines 46-62
- Alterations to the abstract were made to provide a more definitive and clear conclusion statement. – Lines 20-24
- There are no objections to the preparation of the research, careful planning preceded the studies. Unfortunately, the communication of basic data is very limited, and it would be advisable to supplement them.
- Methods and figures were edited to provide a more clear and concise description of the ranges and data to better communicate the data – Lines 177-186
- Since NIRS techniques primarily allow the study of changes, it would have been good to know, for example, the resting laboratory values of the studied properties (Hb, HHB, satO2, etc.).
- Resting values of HHb (arbitrary units) and StO2 (%) were added to the manuscript. – Line 175
- The range values of the dynamometric data could provide information about the properties of the sample examined.
- Minimum and maximum values were added to participant demographics table to better describe the sampled population.
- Based on the sample and method used, it cannot be stated that there are physiological differences in the oxygen supply systems of the two genders. This may be partly true, but stronger contributors should also be assumed, such as ST%, individual capillarization status, relationship between individual muscle mass and physical performance, etc
- Conclusion section was edited to provide less proposed confirmation on exact cause of muscle oxygenation differences between males and females – Lines 378-385
- It is worth mentioning that the applied loads do not specifically count on aerobic sources, it is difficult to give a correct answer regarding the role of energy supply systems in these contractions (see lactate concentration, HR values).
- The inability to confirm the specific energy supply systems was added as a limitation for this study. Lines - 365-369
Additionally:
- FD replaced ME throughout manuscript.
- Body mass replaced weight throughout manuscript.
Round 2
Reviewer 2 Report
Comments and Suggestions for Authors
I accept the authors' answers.
I think the revised manuscript has emproved in quality